# Collagen I Promotes Adipocytogenesis in Adipose-Derived Stem Cells In Vitro

**DOI:** 10.3390/cells8040302

**Published:** 2019-04-01

**Authors:** Nadja Zöller, Sarah Schreiner, Laura Petry, Stephanie Hoffmann, Katja Steinhorst, Johannes Kleemann, Manuel Jäger, Roland Kaufmann, Markus Meissner, Stefan Kippenberger

**Affiliations:** Department of Dermatology, Venereology and Allergy, Johann Wolfgang Goethe University, Theodor-Stern-Kai 7, D-60590 Frankfurt/Main, Germany; Nadja.Zoeller@kgu.de (N.Z.); S_Schreiner@gmx.de (S.S.); Laura.Petry@kgu.de (L.P.); Stephanie.Hoffmann@kgu.de (S.H.); katja.Steinhorst@kgu.de (K.S.); Johannes.Kleemann@kgu.de (J.K.); Manuel.Jaeger@kgu.de (M.J.); kaufmann@em.uni-frankfurt.de (R.K.); Markus.Meissner@kgu.de (M.M.)

**Keywords:** adipose-derived stem cells, adipocytes, differentiation, collagen I, adiponectin, integrins, discoidin domain-containing receptor, ageing, subcutaneous fat

## Abstract

A hallmark of ageing is the redistribution of body fat. Particularly, subcutaneous fat decreases paralleled by a decrease of skin collagen I are typical for age-related skin atrophy. In this paper, we hypothesize that collagen I may be a relevant molecule stimulating the differentiation of adipose-derived stem cells (ASCs) into adipocytes augmenting subcutaneous fat. In this context lipogenesis, adiponectin, and collagen I receptor expression were determined. Freshly isolated ASCs were characterized by stemness-associated surface markers by FACS analysis and then transdifferentiated into adipocytes by specific medium supplements. Lipogenesis was evaluated using Nile Red staining and documented by fluorescence microscopy or quantitatively measured by using a multiwell spectrofluorometer. Expression of adiponectin was measured by real-time RT-PCR and in cell-free supernatants by ELISA, and expression of collagen I receptors was observed by western blot analysis. It was found that supports coated with collagen I promote cell adhesion and lipogenesis of ASCs. Interestingly, a reverse correlation to adiponectin expression was observed. Moreover, we found upregulation of the collagen receptor, discoidin domain-containing receptor 2; receptors of the integrin family were absent or downregulated. These findings indicate that collagen I is able to modulate lipogenesis and adiponectin expression and therefore may contribute to metabolic dysfunctions associated with ageing.

## 1. Introduction

Adipose-derived stem cells (ASCs), residing within fat tissue, display multilineage plasticity, which makes them interesting for regenerative medicine. Minimal invasive procedures allowing liposuction facilitate access, particularly to superficial adipose tissue. After purification and plating, the fate of native ASCs can be directed by specific culture conditions. As a result, different phenotypes can be induced that are characteristic for adipocytes [1,2], fibroblasts [3], endothelial cells [4], osteoblasts [5], chondrocytes [6], cardio-myocytes [7], neural-like cells [2], hepatocytes [8], pancreatic cells [9], and others.

Skin ageing is triggered by intrinsic and extrinsic factors, leading to a massive remodeling of the extracellular matrix (ECM) [10,11]. In this study, we hypothesize that the age-related decrease of ECM molecules also has an impact on the adipogenic differentiation potential in ASCs. In this context, we focus on the role of collagen I, which is the most prominent constituent of ECM molecules distributed throughout the interstitium, making up to 90% of the total connective tissue. Besides providing mechanical rigidity, collagen fibrils also act as guidance structures for contacting cells. Among cellular collagen receptors, the surface receptors of the integrin family are most prominent. They are composed of noncovalently associated α and β subunits forming a heterodimer that recognizes specific amino acid motifs within collagen molecules, hereby initiating intracellular signaling cascades. Besides integrins, two receptor tyrosine kinases, namely discoidin domain receptors 1 and 2 (DDR1 and DDR2), have also been identified to be activated upon collagen binding [12,13]. Particularly, DDR2 is characteristic for cells of mesenchymal origin, such as fibroblasts and smooth muscle cells [14]. 

In order to characterize adipogenic differentiation, we put particular emphasis on lipogenesis by measuring the lipid content. Moreover, another characteristic of adipocytes is the signature pattern of secreted factors that have been collectively termed ‘adipokines’. Prominent among those are leptin, adiponectin, resistin, and visfatin, as well as cytokines and chemokines, such as tumor necrosis factor-α and interleukin-6 [15]. It is assumed that these factors contribute to a subinflammatory state that triggers the development of many chronic obesity-correlated diseases. In this present paper, we focus on adiponectin, a multifaceted adipokine, known as a modulator of inflammation [16,17]. 

The data presented in this paper show that the contact of ASCs to collagen I is a relevant factor in adipogenic differentiation as characterized by lipogenesis and adiponectin secretion. It could be speculated that collagen synthesis under physiological conditions might control the fate of ASCs.

## 2. Materials and Methods

### 2.1. Ethics Statement

This study was conducted according to the Declaration of Helsinki Principles and in agreement with the Local Ethic Commission of the faculty of Medicine of the Johann Wolfgang Goethe University (Frankfurt am Main, Germany). The Local Ethic Commission waived the need for consent.

### 2.2. Isolation and Characterization of ASCs

Isolation and initiation of human ASC cultures were performed as described [18]. As source served abdominal subcutaneous fat tissue derived from plastic surgeries, generously provided by Dr. Ulrich Rieger (Klinik für Plastische und Ästhetische Chirurgie, Wiederherstellungs und Handchirurgie, Markus Krankenhaus, Frankfurt/Main, Germany). The fat tissue was placed in PBS with 2% penicillin/streptomycin (Biochrom, Berlin, Germany) incubated overnight (4 °C). On the next day, skin and blood vessels were mechanically removed by scissors and forceps. Small pieces, with approximately 5 mm lengths, were given to a collagenase type I solution (Worthington, Lakewood, USA) and incubated for 3 h at 37 °C. Cell debris was discarded by filtration through sterile gauze. Then, the cell suspension was centrifuged (400× *g*, 6 min, 4 °C), the cell pellet resuspended in medium, and passed through a cell strainer (70 µm, Greiner, Frickenhausen, Germany). Next, cells were separated by density filtration using a Biocoll solution with a specific density of 1.077 g/mL. After another centrifugation (400× *g*, 30 min, 4 °C), ASCs were isolated from the opaque interphase and seeded in DMEM supplemented with 1% UltroSerG (Pall, Dreieich, Germany) and 1% penicillin/streptomycin. The medium was renewed every 3 days.

### 2.3. Flow Cytometry

Phenotypical characterization of ASCs was performed using the BD FACSCalibur and analyzed with the CellQuest software (v. 1.0.1, Becton–Dickinson, Heidelberg, Germany). The cells were trypsinized, placed on ice for 30 min, and treated with the following labeled stemness-associated antigen markers: CD31-, CD34+, CD45–, CD54–, CD90+, CD105+, CD166+, HLA-ABC+, HLA-DR–. CD34-PE, CD90–FITC, and CD105-PerCP were part of the BD Stemflow Kit (Becton Dickinson, 562245). The other antibodies with the indicated specifications were purchased separately: CD31-FITC (R & D, Systems, Wiesbaden, Germany, FAB3567F), HLA-DR-PE (Becton Dickinson, 347401), CD166-PE (Becton Dickinson, 559263), CD34-APC (Becton Dickinson, 555824), CD54-FITC (Beckman Coulter, Krefeld, Germany, PN IM0726U), HLA-ABC-FITC (Becton Dickinson, 555552), and CD45-PE (Becton Dickinson, 555483). Table 1 shows the results from three donors. All experiments shown in this paper were performed with cells until reaching passage 5.

### 2.4. Adhesion Assay

For the adhesion experiments, ASCs were trypsinized and seeded at a concentration of 1 × 10^4^ cells/well onto microtiter plates coated with collagen I (Becton-Dickinson, Heidelberg, Germany). The coating procedure was performed as previously described [19]. Briefly, collagen type I was dissolved with sterile acetic acid and given in the specific vessels (5 mg/cm^2^). After drying, solvent and collagen remnants were removed by washing with PBS. Regular plastic dishes served as controls. After seeding, the cells were allowed to attach for 30, 60, 90, and 120 min at 37 °C. Then, non-anchored cells were removed by two washings with PBS, and the nuclei of anchored cells were stained with the DNA-binding fluorochrome bisbenzimide (2 mg/mL, 20 min, RT). After two washing steps, fluorescence was detected using the CytoFluor multi-well plate reader (Applied Biosystems, Langen, Germany) at 360/460 nm [19,20]. Experiments were performed three times with three triplicates.

### 2.5. Induction of Adipogenesis

ASCs were seeded in 24-well culture dishes at a concentration 5 × 10^4^ cells/well. Adipogenic differentiation was induced by Adi-medium consisting of DMEM supplemented with 10 µM insulin, 0.5 mM 3-isobutyl-1-methylxanthine, 1 µM dexamethasone, 200 µM indomethacine, 1% penicillin/streptomycin solution, and 2% UltroserG [2,18]. 

### 2.6. Detection of Lipids

ASCs were cultured in Adi-medium (or standard medium) on collagen I-coated and non-coated supports for 9, 11, and 15 days. Consecutively, the lipid content was quantitatively detected as described [21]. Briefly, cultures were washed with PBS and then stained with nile red (1 µg/mL, 20 min, 37 °C). After two washing steps with PBS, fluorescence documented by fluorescence microscopy or quantitatively measured using a multiwell spectrofluorometer (Cytofluor, Applied Biosystems, Langen, Germany) equipped with 485/560 nm filters for neutral lipids. Moreover, cell nuclei were stained with bisbenzimide (2 µg/mL, 20 min, 37.4 °C) and the fluorescence, as a measure for cell count, was detected at 360/460 nm. In order to normalize the lipid values to different cell counts, a ratio was formed between nile red and bisbenzimide measurements. The values derived from cells cultured on plastic were set to 100% and all other values were related to that. Experiments were performed in quadruplicate, with 4 parallel determinations for each condition.

### 2.7. Western Blot

Cells cultured as described above were lysed in 100 mL SDS sample buffer (62.5 mM Tris-HCl [pH 6.8], 2% SDS, 10% glycerol, 50 mM DTT, 0.1% bromphenol blue), sonicated and boiled for 5 min, and separated on SDS-polyacrylamide gels. Consecutively, proteins were immunoblotted to a PVDF membrane. The membrane was blocked in blocking buffer (TBS [pH 7.6], 0.1% Tween-20, 5% nonfat dry milk) for at least 3 h at 4 °C followed by incubation with the primary antibody in TBS (pH 7.6), 0.05% Tween-20, and 5% BSA. Bound primary antibodies were detected using anti-mouse IgG-horseradish peroxidase conjugate and visualized with the LumiGlo detection system (Cell Signaling, Frankfurt, Germany). The following primary antibodies were used: Integrin β1 (Santa Cruz, sc-6622), integrin α1 (R & D, MAB5676-SP), integrin α2 (Becton-Dickinson, 611016), integrin α11 (R & D, AF4235-SP), and discoidin domain-containing receptor 2 (DDR2) (R & D, AF2538).

### 2.8. Adiponectin ELISA

Cells were seeded on collagen or plastic supports and exposed to either Adi-medium (or standard medium) for 9, 11, and 15 days. Cell-free supernatants were obtained and assayed for human adiponectin using a commercial ELISA test kit (R & D) according to the manufacturers’ instructions. Briefly, supernatants were placed in microwell plates coated with antibodies against adiponectin. After incubation with a biotin-labeled secondary antibody, a streptavidin horseradish-peroxidase conjugate was added. A colorimetric reaction was exerted by addition of a substrate (tetramethylbenzidine/peroxide), giving rise to a colored product measured at 450 nm in a scanning multiwell spectrophotometer (ELISA reader MR 5000, Dynatech, Guernsey, UK).

### 2.9. Real-Time RT-PCR Analysis

Total cellular RNA was isolated from cells cultured as described above using the ExtractMe Total RNA Kit (Blirt, Gdansk, Poland). After DNase digestion, a total amount of 25 ng RNA was used for first-strand cDNA synthesis using a QuantiTect SYBR Green RT-PCR Kit (Qiagen, Hilden, Germany). Real-time PCRs were performed on a Light Cycler system 2.0 (Roche Diagnostics, Mannheim, Germany). The following primers for adiponectin (accession no. ENST00000320741.6) and PBDG (accession no. ENST00000278715) were used: 

Adiponectin-for: 5′-TGTGGTTCTGATTCCATACCAG-3′

Adiponectin-rev: 5′-CGGGCAGAGCTAATAGCAGTA-3′

PBDG-for: 5′-CCATGTCTGGTAACGGCAAT-3′

PBDG-rev: 5′-GTCTGTATGCGAGCAAGCTG-3′

The relative expression of transcripts was determined using the 2-∆∆CT method [22].

### 2.10. Statistics

All data are presented as mean values ± standard deviation. Statistical significance of the data was calculated by Wilcoxon-Mann-Whitney-U-test (BIAS, Frankfurt, Germany). Data sets were statistically compared as indicated in the graphics.

## 3. Results

Although most cell cultures were carried out on uncoated plastic dishes, it is known that extracellular matrix molecules have impacts on cell physiology. Therefore, we initially tested the adherence of ASCs held in standard medium to collagen I in comparison to uncoated plastic (Figure 1). Representative photographs of time-dependent adhesion are shown in Appendix A. It was found that ASCs in suspension feature a significant faster adherence to collagen I-coated substrates in the first 30 min. In the further course, the cell count adherent to either plastic or collagen I was similar in the observed time span. These results indicate that naïve ASCs also express adhesion molecules with avidity to collagen I. 

Furthermore, we investigated the impact of adhesion to collagen I on cell morphology and differentiation (Figure 2). Cells were seeded on supports coated with collagen I or plastic for control and then cultured for 19 days in either standard medium (DMEM) or adipogenic medium (Adi-medium). On plastic (Figure 2A), ASCs feature a domed cell center with filigree dendritic plasma extrusions in the periphery. These dendritic branches are less pronounced in cells on collagen I (Figure 2B). When cells were held in adipogenic medium, the cell morphology changed massively. Cells form a large interconnected network with many extended cell branches reminding on a network of nerve cells (Figure 2C,D). Moreover, cells accumulate tightly packed vesicles in the cell center, which putatively represent lipid droplets. In order to verify the nature of these droplets, a nile red staining was performed (Figure 2E–H). Positive nile staining was found around the cell nucleus, stained in blue with bisbenzimide, indicating the presence of lipids. Of note, on the photographs, the expression of lipids seems to be more distinct in cells cultured on collagen I.

In order to validate this first impression, nile red staining was quantified at different time points (9, 11, 15 days) by fluorometric means as described. Figure 3 shows the content of neutral lipids of ASCs in dependence to culture medium (DMEM vs Adi-medium) and cell support (plastic vs collagen I). The value determined for ASCs cultured in DMEM on plastic was set to 100%. In these cells, which are absent of lipid droplets, nile red stains showed lipid-containing membranes only. All other measured values were related to this. It was found that cultivation on collagen I led to a slight, but significant increase in lipid content at all examined time points in DMEM (black bars). Moreover, the change to Adi-medium induced the expected massive and significant increase in lipids in cells on plastic and collagen I (striped bars). Of note, cultures on collagen I showed an increase in lipids compared to their counterparts held on plastic. In sum, these results indicated a lipogenic effect of collagen I in ASCs.

Next, the release of adiponectin, a prototypical adipokine, was evaluated in cell-free supernatants by ELISA after 9, 11, and 15 days (Figure 4A–C). ASCs cultured in DMEM showed no significant release of adiponectin in the period under observation regardless of whether they were cultured on plastic or collagen I (black bars). Changing the medium to Adi-medium induced a massive release of adiponectin (striped bars). Comparison of both cell substrates shows that the measured levels of adiponectin on collagen I were significantly reduced. 

To learn more about the level of this regulation, mRNA transcripts of adiponectin were quantified by real-time RT-PCR analysis (Figure 5). As found for the adiponectin release, almost no adiponectin mRNA expression was found in ASCs held in DMEM (black bars). In contrast, massive induction was initiated by changing the medium to Adi-medium (striped bars). Similar to the measured protein levels, this induction was gradually diminished by cultivation on supports coated with collagen I. The measured Ct values ranged from ca. (circa) 29 for non-differentiated to ca. 22 for differentiated cells. The Ct values for reference gene (PBGD) expression were ca. 27 independent from culture conditions.

Our results demonstrate that collagen I has an impact on the cell physiology in ASCs. Therefore, the expression of collagen receptors was determined by western blot analysis (Figure 6). Surface receptors of the integrin family form heterodimers composed of α and β subunits, which convey substrate specific recognition. For the recognition of collagens, the α1, α2, α11, and β1 subunits are relevant. Positive controls for the detection of α1 and α11 were protein lysates derived from normal human fibroblasts; for α2 and β1, lysates from HaCaT keratinocytes were used. It was found that ASCs in DMEM produce distinct amounts of α1 on plastic as well as on collagen I (Figure 6A). A change to Adi-medium significantly reduced this expression. Likewise, α11 was massively reduced in ASCs cultured in Adi-medium. Interestingly, the basal level in ASCs held in DMEM was in the range of the positive control. Integrin α2 was only present in the positive control. The β1 subunit was even more pronounced than in the positive control, particularly in ASCs held in DMEM. A shift to Adi-medium caused a decrease in β1 expression. Moreover, DDR2, another collagen receptor present in mesenchymal cells, was investigated (Figure 6B). Here, a moderate expression in ASCs cultured in DMEM was found. Of note, a shift to Adi-medium caused a profound upregulation. Lysates derived from fibroblasts served as a positive control.

## 4. Discussion

Collagen I, as the most abundant ECM molecule, is produced by fibroblasts. It is known to provide a structural scaffold for cell attachment with impacts on tissue organization and tissue homeostasis by affecting cell growth, motility, viability, and differentiation [23]. The omnipresence of collagen in the mesenchym makes this molecule interesting as a potential trigger factor determining the cell’s fate. It is known that the amount of collagen decreases during ageing with implications to the above mentioned physiological parameters [24]. An imbalance of the production and degradation leads to the accumulation of fragmented collagen molecules on which fibroblast cannot efficiently attach to. The absence of a mechanical load, which is a trigger factor for proliferation [25], marks the entry to a self-perpetuating detrimental cycle, accelerating the decrease of collagen [24]. Moreover, another hallmark of ageing is the redistribution of fat, with a decrease of peripheral subcutaneous fat [26]. It is an interesting issue if both observations are somehow functionally linked. ASCs residing in subcutaneous fat tissue have the potential to transdifferentiate into adipocytes, serving to compensate the age-dependent volume loss. In this paper, we showed that ASCs produce more lipids, a marker for adipogenic differentiation, when cultured in collagen coated dishes. This effect was measured both in cells cultured in standard medium containing no differentiation factors, and also in cells cultured in Adi-medium, described to promote adipogenic differentiation. Against this background, the stimulation of collagen synthesis by topical retinoic acid [27,28,29] or carbon dioxide laser resurfacing [30] may also support the differentiation of ASCs into adipocytes. Furthermore, injectable preparations containing animal collagen, which are widely used in cosmetic medicine as injectable fillers to increase lip volume, to minimize wrinkles, or to correct post-acne and traumatic scars [31], impact this differentiation process. 

In addition to lipogenesis, the expression of adiponectin was also examined, a fat-derived hormone that is predominantly produced by adipocytes [32]. Corroboratively, ASCs in standard medium produce no adiponectin while cultivation in Adi-medium leads to a massive induction, indicating successful transdifferentiaton to adipocytes. Although adiponectin has pleiotropic metabolic effects, it is mainly known for its action as an insulin sensitizer with anti-apoptotic and anti-inflammatory properties [33,34,35]. Interestingly, the presence of collagen suppressed the expression of adiponectin. In humans, an inverse correlation between fat mass and adiponectin plasma levels was observed [36,37]. In this context, the inhibitory effect of collagen may represent a countermeasure in response to the massive induction of lipogenesis. 

After we learnt that collagen impacts on metabolic parameters in ASCs, we looked at the potent receptors conveying such a signal, namely integrins and DDR2. In higher vertebrates, there are 18 α subunits and 8 β subunits, which form 24 distinct integrins. Specific for collagen binding are the following heterodimers: α1β1, α2β1, α10β1, and α11β1 [38]. Therefore, in western blot analysis, the expression of these integrins were investigated. An inverse regulation between lipogenesis/adiponectin expression and the expression of α1, α11, and β1 was found (α2 was not detectable in ASCs independent from culture conditions). In contrary, DDR2 was massively induced in ASCs when transdifferentiated to adipocytes. These findings suggest a contribution of this receptor in the collagen I-induced regulations of lipogenesis and adiponectin expression. 

In sum, the presented results identified collagen I as a new regulator of adipogenic differentiation of ASCs. These findings have implications for the understanding of changes in age-related fat metabolism. 

## Figures and Tables

**Figure 1 cells-08-00302-f001:**
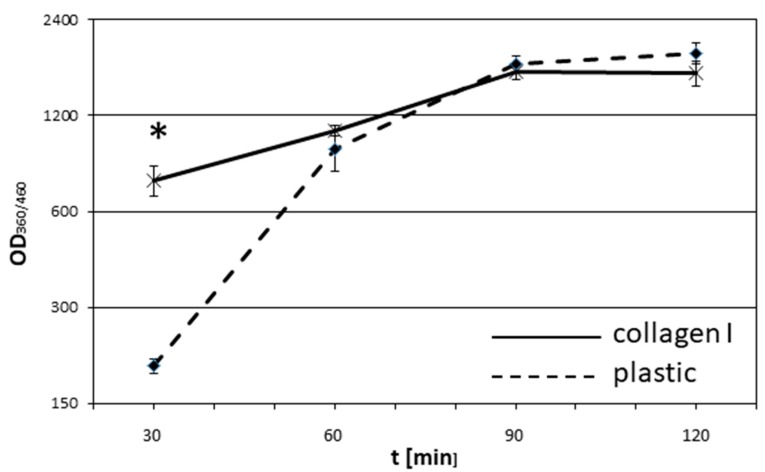
Quantitative adhesion of ASCs on plastic vs collagen I. ASCs were plated at a density of 1 × 10^4^ cells per well into microtiterplates either non-coated or coated with collagen I. After 30, 60, 90, and 120 min, non-adherent cells were discarded and the remaining cells were stained with bisbenzimide. Fluorescence, as a measure for adherent cells, was quantified using a CytoFluor multi-well plate reader at 360/460 nm. Each point represents the mean of five independent experiments. The standard deviations are indicated. * *p* < 0.05. The whole experiment was repeated three times with similar results.

**Figure 2 cells-08-00302-f002:**
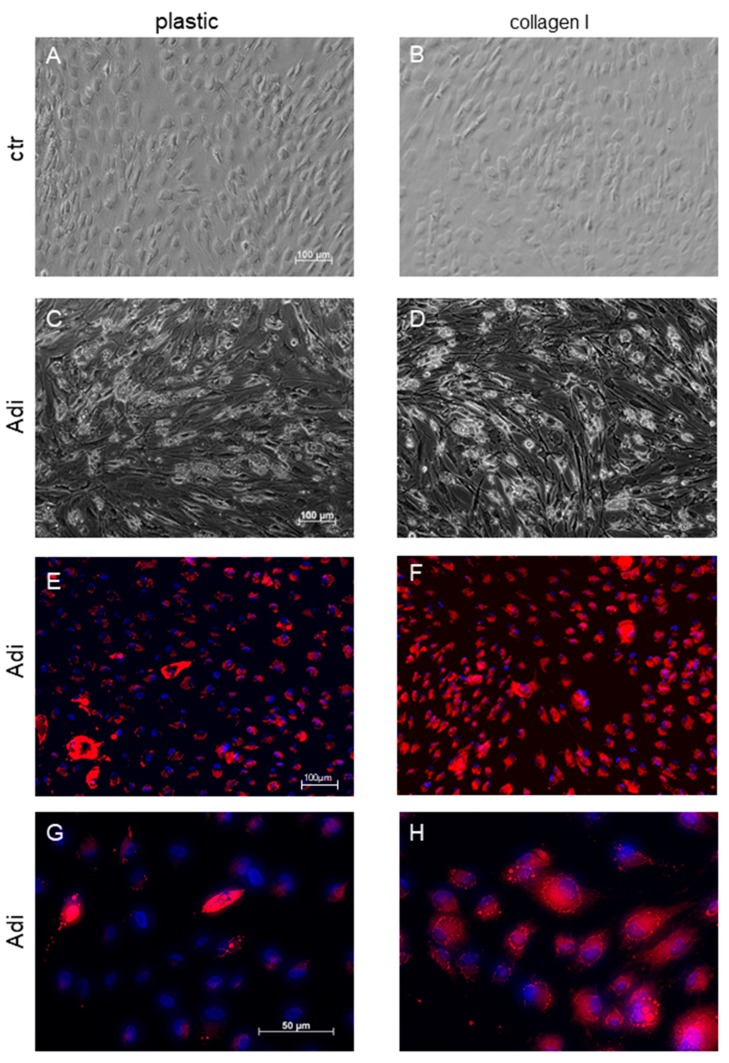
ASCs cultured on plastic or collagen I. (**A**,**B**) show the cell morphology after cultivation in ASC standard medium (ctr) for 19 days, (**C**,**D**) after cultivation in adipogenic medium (Adi). (**E**–**H**) display cellular lipids (red) after cultivation in adipogenic medium for 19 days stained with the fluorescence dye nile red; cell nuclei (blue) are stained with bisbenzimide. Photographs show representative sections.

**Figure 3 cells-08-00302-f003:**
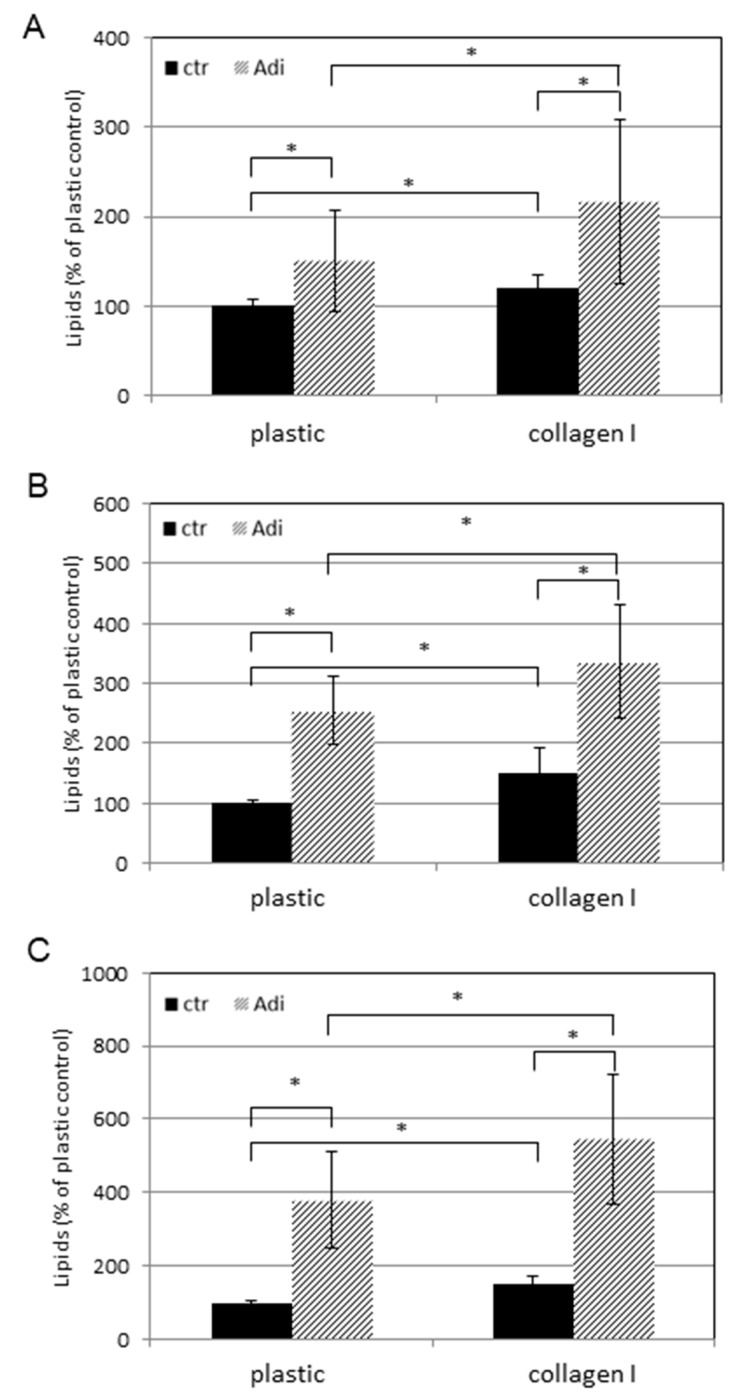
Quantitative lipogenesis of ASCs on plastic or collagen I. ASCs were plated at a density of 1 × 10^4^ cells per well into microtiterplates either non-coated (plastic) or coated with collagen I. One part was held in regular DMEM medium (ctr) and the other in adipogenic medium (Adi). After (**A**) 9 days, (**B**) 11 days, and (**C**) 15 days, cells were stained with nile red and bisbenzimide and the fluorescence was measured (see Materials and Methods). Each point represents the mean of 16 independent experiments. The standard deviations are indicated. Data sets were statistically compared as indicated. * *p* < 0.05.

**Figure 4 cells-08-00302-f004:**
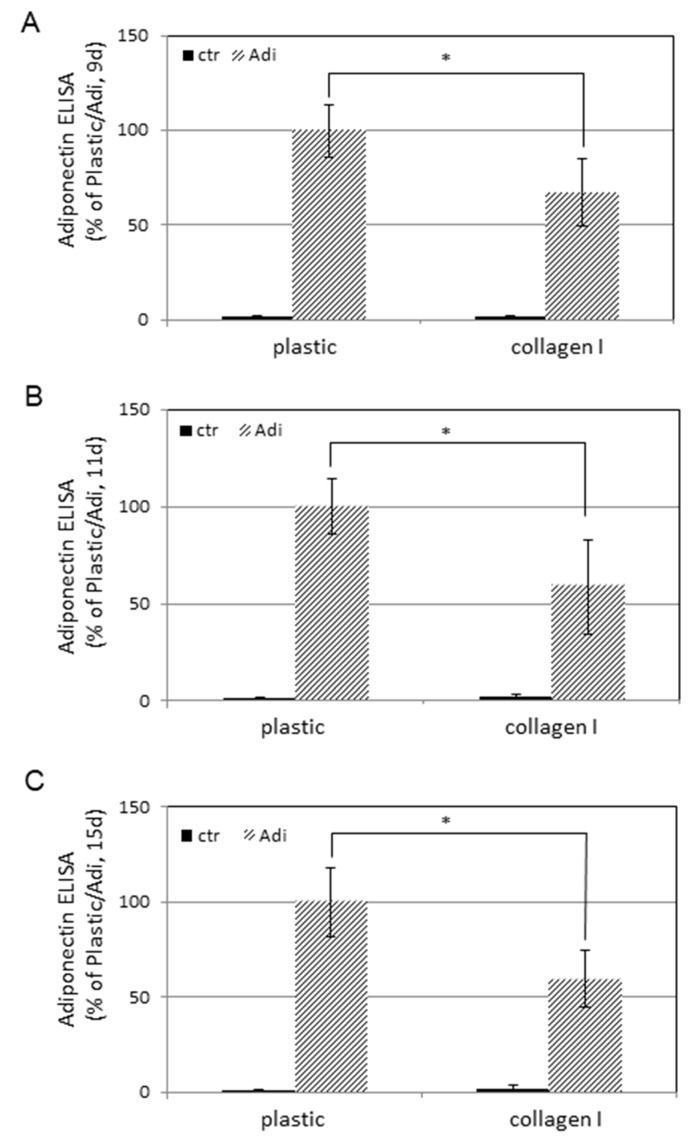
Adiponectin release of ASCs on plastic or collagen I. ASCs were plated at a density of 1 × 10^4^ cells per well into microtiterplates either non-coated (plastic) or coated with collagen I. One part was held in regular DMEM medium (ctr) and the other in adipogenic medium (Adi). After (**A**) 9 days, (**B**) 11 days, and (**C**) 15 days, supernatants were examined for adiponectin by ELISA. Each point represents the mean of 20 independent experiments. The standard deviations are indicated. Data sets of Adi/plastic and Adi/collagen I were statistically compared as indicated. * *p* < 0.05.

**Figure 5 cells-08-00302-f005:**
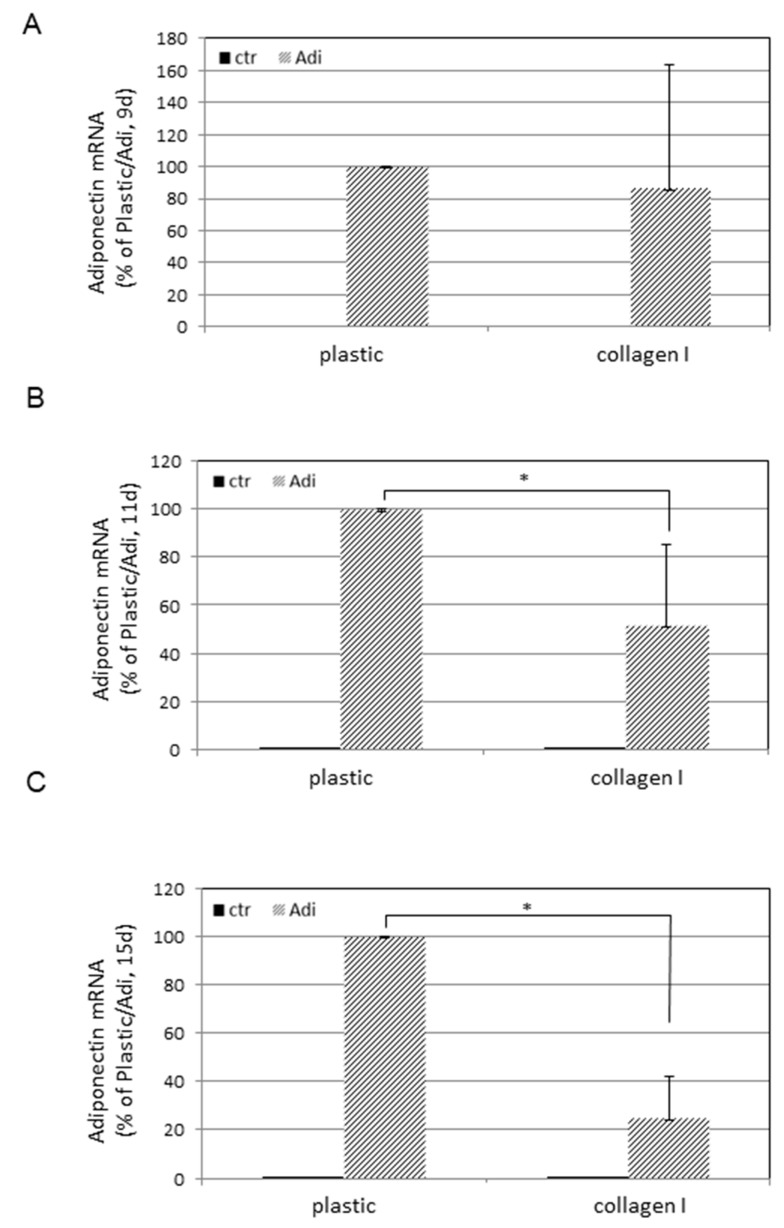
Adiponectin mRNA expression of ASCs on plastic or collagen I. ASCs were plated at a density of 1 × 10^4^ cells per well into microtiterplates either non-coated (plastic) or coated with collagen I. One part was held in regular DMEM medium (ctr) and the other in adipogenic medium (Adi). After (**A**) 9 days, (**B**) 11 days, and (**C**) 15 days, total RNA was extracted and real-time RT-PCR analysis was performed. The values for ASCs in DMEM are less than 1%. Data sets of Adi/plastic and Adi/collagen I were statistically compared as indicated. * *p* < 0.05.

**Figure 6 cells-08-00302-f006:**
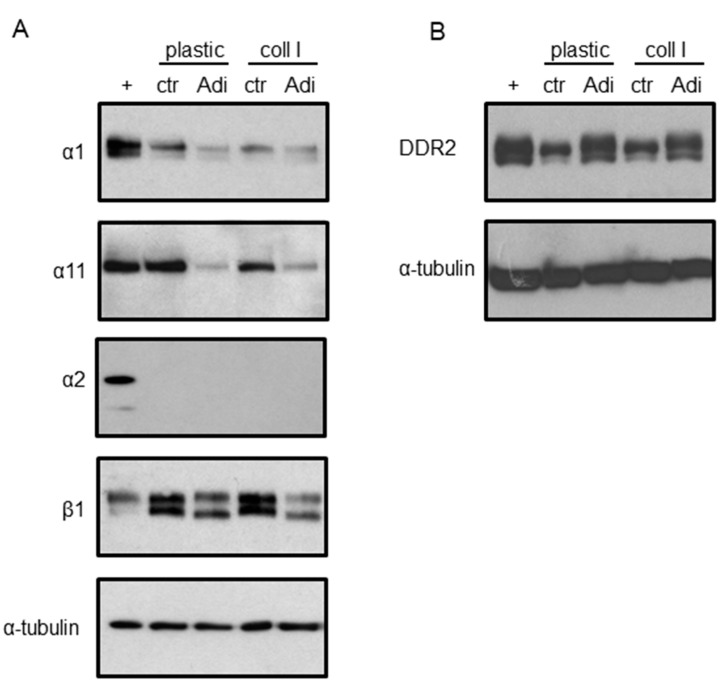
Expression of collagen I receptor molecules. ASCs, plated on regular plastic supports or supports coated with collagen I, were held for 9 days in standard DMEM medium (ctr) or in adipogenic medium (Adi). Protein extracts were subjected to western blot and tested for (**A**) α1, α2, α11, β1 integrin, and (**B**) DDR2. Equal loading was monitored by using antibodies directed against α-tubulin. The blots show representative results (*n* = 3). + indicates positive controls.

**Table 1 cells-08-00302-t001:** Characterization of ASCs.

Surface Marker	Positive Cells [%]		
	Donor 1	Donor 2	Donor 3
CD31/PECAM-1	0.75 ± 0.29	1.30 ± 0.19	0.83 ± 0.02
CD34	53.94 ± 7.09	63.98 ± 13.67	78.00 ± 1.29
CD45	0.85 ± 0.10	1.10 ± 0.13	0.97± 0.11
CD54/ICAM-1	1.36 ± 0.36	1.88 ± 0.57	2.76 ± 1.06
CD90/Thy-1	99.96 ± 0.03	99.88 ± 0.11	99.09 ± 1.39
CD105/Endoglin	91.07 ± 4.92	90.82 ± 11.62	60.17 ± 5.36
CD166/ALCAM	98.93 ± 0.82	97.83 ± 1.80	99.12 ± 0.11
HLA-ABC	99.74 ± 0.17	99.88 ± 0.05	99.48 ± 0.27
HLA-DR	0.78 ± 0.46	0.72 ± 0.11	0.56 ± 0.10

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
