# Peer review of "Collagen I Promotes Adipocytogenesis in Adipose-Derived Stem Cells In Vitro"

_cells, 2019, doi:10.3390/cells8040302_

Round 1
Reviewer 1 Report
This is an interesting study investingating a novel concept, which could explain changes in subcutaneous fat during ageing. The study is well performed and the conclusions drawn from the experiments are valid. I have the following comments and questions.
Major comments:
1. To me, the photomicrographs in Fig. 2 are not convincing. I would have expected to see more and somewhat bigger lipid droplets in the cells by day 19. There are just the odd cells with visible lipid droplets, the other cells only exhibit a faint glow only. In this setting, how long does it take to observe lipid droplets in differentiating adipocytes with e.g. Oil Red O staining? It also struck me that the cells growing on plastic failed to properly differentiate into adipocytes, whilst I would have thought that this medium is a conventionally used adipogenic medium. Please could the authors comment on this?
2. For RT-qPCR reactions, it is not described how exactly the normalizing gene (PBDG) has been chosen. Did the authors check more than one normalizing gene before choosing the one they used? Please could the authors comment on this.
3. For section 2.4 in the Materials and Methods, the coating procedure has not been described in detail (e.g. concentration, incubation time, etc).
4. In Fig. 6, why are the bands for tubulin appear much darker/stronger for DDR2 than for the integrins?
5. Are the authors aware of data (either own or published) with regards to the number of ASCs in young vs. aged adipose tissue?
6. I don’t think that the term ‘transdifferentiation’ is correctly used in the manuscript, such as in line 59. To my knowledge, transdifferentiation is a process in which one mature somatic cell transforms into another mature somatic cell, without undergoing an intermediate pluripotent state or progenitor cell type. In this case, the authors studied the differentiation of ASCs into adipocytes. Therefore, please replace the occurrences of the term ‘transdifferentiation’ to ‘differentiation.’
7. In line 73, the authors state that adipose tissue, after having incubated it at 4°C ON, was separated from blood vessels - please may I ask how this was done? Was it a simple mechanical separation, or did it involve enzymatic digestion as well?
Minor comments:
1. In line 92, please replace the word ‘trypsinated’ for ‘trypsinized.’
2. In line 102, please change the concentration value from 0.5 x 10⌃5 to 5 x 10⌃4.
3. Please correct the spelling of western blot throughout (not capital W).
Author Response
Answer to reviewer 1.
The authors like to thank the reviewer for reading the MS and for constructive critique.
1. ASC grown in adipogenic medium (Adi) show a marked production of lipids which can be seen both, by light microscopy (Fig. 2 C, D) and by qualitative Nile Red staining (2 E - H) – and, moreover by quantitative Nile Red staining (Fig. 3 A – C). This documents adipogenic differentiation by Adi. Fig. 2 E, F presents photographs as an overview to show that most of the cells have been differentiated. Because of a strong Nile Red staining the display of single lipid droplets is somehow outshined. Therefore, higher magnifications have been added (Fig. 2 G,H). In the beginning of this study also other lipid staining dyes were tested giving similar results. We decided to work with Nile red because of economic reasons and convenient handling.
2. Yes, GAPDH was also tested for normalization in RT-qPCR. But as Ct-values were not within the same range as our target gene we took PBDG which is expressed in the same range.
3. The coating procedure was added to the Materials and Methods section (pls see Materials and Methods, Adhesion Assay).
4. For western blot analysis equal amounts of proteins were separated. However, the intensity strengths also dependent from the exposure time. This might lead to different intensities particularly for tubulin. For better discrimination the image was replaced by a shorter exposed film.
5. In our experiments we found no relation to donor age in regard to ASC number.
6. Thank you for pointing us to the difference between differentiation and transdifferentiation. Accordingly, the term “transdifferentiation” was replaced by “differentiation” throughout the text.
7. The isolation procedure of ASCs from fat tissue was clarified in the text. Skin and blood vessels were mechanically removed by scissors and forceps followed by an enzymatic digestion as described (pls see Materials and Methods, Isolation and characterization of ASCs).
Minor comments:
1. “trypsinated” was replaced by “trysinized”.
2. “0.5 x 105” was replaced by “5 x 104”.
3. “Western blot” was changed to “western blot”
Reviewer 2 Report
Zoeller and colleagues describe the role of Collagen I in directing adipose mesenchymal stem cell towards adipogenic differentiation. The paper is clearly presented although some major points have to be addressed in more details:
- M&M: please indicate the code for all antibodies used througout the study; further, for flow cytometry please indicate the conjugation;
- Table 1: in the three ASC isolates, a remarkable amount of CD34 is detected. There is an ongoing debate about presence of CD34 in cultured ASC, with strong evidences that this molecule decrease its abundance after few passages and population doublings. In M&M it is stated that flow cytometry has been performed untill passage 5, therefore CD34 presence looks like strange if passage was 4-5. Authors should describe the presence of CD34;
- M&M: in the paragraph describing adhesion assay, please specify cell number per square cm;
- Line 163: "solubilized" should be replaced with "in suspension" or similar;
- Fig. 1: in the caption of the figure, it is stated that the shown experiments have been performed 3 times with similar results. Does this mean that the graph refers to only one experiment? Or is it the mean value of all 3 experiments merged?
- Fig. 2: in the text, it is mentioned that ASC show dendritic branches and that with Collagen they are less pronounced. Authors should quantify this value;
- Fig. 3: In the lipid quantification, a value for ASC is shown also without adipogenic induction. Authors should explain the origin of this value in steady state condition and how Nile red is specific;
- Fig. 4: authors state that adiponectin release without induction is not significant. This may sound that ELISA was not able to give a reliable readout. Therefore, how is possible to have ratios with something that is not detectable? Authors have to present the data indicating the concentration of the adiponectin and also the range of detection of the kit used;
- Fig. 5: again, authors state that adiponectin transcript was almost undetectable without adipogenic stimulation in qRT-PCR. Therefore, how is it possible to have reliable quantifications? Authors have to indicate Ct values and if they are higher than 30, mention that a reliable quantification is not possible;
- Fig. 6: a supplementary table summarizing the results of WB analysis would help the reader;
- Discussion: the whole paper is based on the assumption that with age the amount of collagen decreaes and therefore, if collagen triggers ASC differentiation into fat, also amount of subcutaneous adipose tissue may reduce. Therefore, a deeper characterization of ASC and donors used in the study is mandatory. Authors have to describe donors indicating sex, age and, if present, pathologies. Also, the presence and structure of collagen in the adipose tissue may vary depending on the donor, therefore index as BMI must be included. To corroborate presented data, authors should perform qRT-PCR and WB on ASC readily isolated from adipose tissue of young and aged donors and from thin or overweighted donors, and not only after prolonged cell culture.
Author Response
Answer to reviewer 2.
The authors like to thank the reviewer for reading the MS and for constructive critique.
1. As requested the specifications of antibodies used were added in the M&M section (pls see under “Flow Cytometry” and “Western Blot”).
2. We agree that there is debate of the relevance of certain stemness markers in ASC. In general CD34+ is still considered as a marker of ASCs (Mildmay-White, A. and W. Khan (2017). "Cell Surface Markers on Adipose-Derived Stem Cells: A Systematic Review." Current Stem Cell Research & Therapy 12: 484-492) although, there are donor-specific differences in expression documented (e.g. Baer, P. C., et al. (2013). "Comprehensive phenotypic characterization of human adipose-derived stromal/stem cells and their subsets by a high throughput technology." Stem Cells Dev 22: 330-339) which is in concert with our findings. The characterization of the ASCs in our experiments was performed instantly after the initial cell culture at passage 1. This characterization was not repeated. The experiments were shown in this paper were performed with cells until passage 5. For clarification the following sentence was added: “All experiments shown in this paper were performed with cells until reaching passage 5” (pls see M&M, Flow Cytometry).
3. The adhesion assay displays adhesion in a semiquantitative manner. It is based on the principle that the amount of nuclei staining, measured by fluorescence intensity after incubation with the DNA dye bisbenzimide, correlates with the cell count. This method delivers robust and reliable data as documented (e.g. Kippenberger, S., et al. (2004). "Ligation of the beta4 integrin triggers adhesion behavior of human keratinocytes by an "inside-out" mechanism." J Invest Dermatol 123: 444-451). Absolut cell counts were not evaluated. A supplementary file was added showing adherent cells on plastic and collagen I in a time dependent manner after staining with bisbenzimide (pls see. Fig. S1). A referring text was added in under Results (“Representative photographs of time-dependent adhesion are shown in Fig S1.”)
4. The term “solubilized” was replaced by “in suspension”.
5. Fig. 1 shows data of an exemplary experiment with three independent parallel determinations. This experiment was repeated 3 times with qualitatively similar results.
6. Fig. 2 shows morphological characteristics of ASCs in regard to different cell supports (plastic vs collagen I) and culture conditions (ctr medium vs Adi medium). In the results section we try to describe by words the evident characteristics knowing full well that a picture tells more than words. I think you agree with us, that cells on collagen I are somewhat more branched than their counterparts on plastic. It is difficult and technically unsatisfying to put this into a quantitative presentation.
7. This is an interesting point. Although, ASCs in control medium show no lipid droplets staining with nile red features a low background signal. Besides lipid droplets nile red stains also other lipids prominently those of membranes leading to a low background signal which is similar in all tested cells. For clarification the following sentence was added: “In these cells which are absent of lipid droplets nile red stains lipid-containing membranes only” (pls see Results).
8. Due to a misleading axis label there was a misunderstanding about what was measured in the adiponectin ELISA (Fig. 4) and qRT-PCR (Fig. 5). All measured values were related to cells on plastic held in Adi medium. The misleading label “% of Plastic/Adi” was replaced by “% of Plastic, Adi”.
9. As aforementioned the axis label in Fig. 5 was corrected. As requested information about the measured Ct-values was added: “The measured Ct values ranged from ca. 29 for non-differentiated to ca. 22 for differentiated cells. The Ct values for reference gene (PBGD) expression were ca. 27 independent from culture conditions” (pls see Results).
10. We agree it is desirable to have more information about donors providing the fat tissue used in our experiments. Because of different reasons, with data protection being the most important, we can provide only sparse information. Most of the material was derived from women with a median age about 50 years. The surgeries were motivated by aesthetical reasons. Concomitant diseases were not known to us. We observed in our experiments only quantitative but no qualitative differences between the different samples. Important to us was that the fat tissue used originates from classical obese patients and not from lipedema, which is a hormone-related disease.
Round 2
Reviewer 2 Report
Authors answers increased clarity of the text and results. The only concern is about flow cytometry characterization. They stated that it was performed right after initial cell culture at passage 1 but the experiments were performed up to passage 5. Flow cytometry for CD34 at this passage should be presented.
Author Response
Q: Authors answers increased clarity of the text and results. The only concern is about flow cytometry characterization. They stated that it was performed right after initial cell culture at passage 1 but the experiments were performed up to passage 5. Flow cytometry for CD34 at this passage should be presented.
Response: As indicated in the paper the stemness of ACSs was characterized by a set of well-established markers at passage 1. As others we restricted our experiments to early passages (until P5) to avoid individualizing effects. Also relevant in this context, fetal bovine serum, known as differentiation-promoting agent, was substituted by a serum-free component (1% UltroSerG). In regard to CD34 it has already been addressed by other studies, that this stem cell-associated marker has a peak levels in very early passages (P0, P1) but remained present, although at reduced levels, throughout higher passages (pls see: Mitchell, J. B., et al., 2006, "Immunophenotype of human adipose-derived cells: temporal changes in stromal-associated and stem cell-associated markers." Stem Cells 24: 376-385). It remains to be seen if CD34 serves as a unique identifier of ASCs (ibidem).